# Informed Exploration via Generative Modeling

## Abstract

Conventionally trained neural networks excel at prediction but often struggle to model uncertainty in their own predictions. We explore this challenge in a meta-learning bandit decision-making problem for news recommendations; this setting require decision-making algorithms to incorporate pretrained language models to process text data for the best performance. We present a scalable approach to Bayesian uncertainty quantification by posing it as a problem of autoregressive generative modeling of future rewards. First, we use historical data on previously released news articles to pre-train a generative model to predict sequences of future potential rewards. At inference time, our algorithm makes decisions based on limited previous rewards and autoregressively generated future rewards. Far from a heuristic, we synthesize insights from the literature to show our method is a novel implementation of Thompson (posterior) sampling, a prominent bandit algorithm. We prove our pretraining loss directly controls online decision-making performance, and we demonstrate our framework on a news recommendation task where we integrate end-to-end fine-tuning of a pretrained language model to process news article headline text to improve performance.

## 1 Introduction

Settings with recurring interactions between humans and an AI decision-making system are becoming increasingly prevalent. Some examples include systems that recommend to users what to buy, and settings in digital health and online education that encourage people towards behaviors aligned with their goals. In these domains, there is a need for decision-making algorithms that can leverage neural networks. Although neural networks excel at making predictions based on intricate text and visual inputs, effective sequential decision-making requires the ability to comprehend uncertainty—to recognize when the model is unsure in its predictions and what additional information might benefit future decision-making. Here neural networks often struggle (Gawlikowski et al., 2023).

In this work we focus on a particular problem setup that is stylized, but which we believe captures key challenges of real-world human-interactive decision-making problems. As depicted in Figure 1 we consider a news recommendation problem where each day, a set of news articles is released. These released articles have no associated historical recommendation data that can be used to learn about those particular articles. However, a neural network model with a large language model (LLM) component can be trained using historical recommendation data on articles from previous days to predict how likely users are to click on a new article if recommended, based on the article's text. At the same time, the neural network's predictions based on the LLM are not perfect, and it is important for the algorithm to know how much to "trust" these predictions. Thus, the challenge is designing a decision-making algorithm that is able to (i) incorporate the LLM's initial prediction/beliefs, (ii) sharpen its beliefs as it recommends that article throughout the day, and (iii) use the LLM's belief to make recommendation decisions in a way that balances exploration and exploitation.

We frame this problem as a meta-learning multi-armed bandit problem. Effective algorithms must be able to learn from user interactions within each day, as well learn from historical recommendation data collected from previously released news articles. This problem requires algorithms to grapple with uncertainty: with more data we could more precisely learn the effectiveness of a particular action. Is it worth taking that action to learn more? Thompson (posterior) sampling (Russo et al., 2020) manages this trade-off by taking actions based on samples from the posterior distribution over

Figure 1: **Daily Online Decision-Making Problem.** Every day a set of news articles is released and the system must recommend these articles to users. The decision-making algorithm incorporates an LLM that reads the text and provides an initial belief/prediction each article's performance if recommended. This belief is refined as the algorithm interacts with users and collects more data.

mean rewards. To scale this idea to modern decision making problems, a variety of heuristics have been proposed to approximate this with neural networks (Riquelme et al., 2018; Snoek et al., 2015; Osband et al., 2018; 2023; Qin et al., 2022b; Lee et al., 2023b; Weitong et al., 2021).

We propose a markedly different approach to Thompson sampling that does not explicitly model unknown latent variables. Rather, our algorithm focuses on modeling *missing rewards* (including future rewards) and considers missing rewards the source of the decision-maker's uncertainty. Our meta-learning procedure operates by using historical data to implicitly learn a Bayesian model, by pre-training an autoregressive generative sequence model to predict future rewards. At inference/decision time, we use the sequence model to autoregressively sample values of missing outcomes to make decisions; additional training of this model is not required. We synthesize insights from the literature to explain that this approach is no heuristic: It is a novel implementation of proper Thompson sampling (with approximate empirical Bayes) if the autoregressive model accurately reflects the data distribution. In designing this algorithm, we re-frame a problem of sequential decision-making and uncertainty quantification as a problem of training an autoregressive generative model to impute missing outcomes. A summary of our contributions is as follows:

1. **Conceptual.** We formalize a meta-bandit problem setting (based on the news recommendation problem) that both motivates and crystallizes insights connecting posterior sampling and generative sequence modeling of missing rewards (Section 2).

2. **Algorithm.** We connect these insights to decision-making and use them to derive a new, scalable implementation of Thompson sampling that can easily incorporate neural networks (Section 3). Our approach obviates the need to use complicated and/or heuristic posterior approximation methods (like MCMC or Bayesian neural networks) when incorporating neural network models. Our procedure implicitly learns the Bayesian model via pre-training by minimizing a sequence prediction loss via gradient descent.

3. **Theory.** We provide formal links between interactive decision-making and sequence prediction, including a novel regret bound that scales with the pre-training loss of the sequence model (Section 4). Our result formally shows that our approach effectively *reduces a challenging sequential decision-making problem to one of learning a sequence prediction model with low loss.*

4. **Experiments.** We demonstrate that our theoretical insights bear out in simulations, and even scale even to the news article recommendation setting where incorporating a pre-trained language model to read the article headline is needed for the best performance. We also find that our sequence pre-training approach implicitly learns a Bayesian model with very accurate uncertainty quantification (credible intervals), which is typically very challenging with neural networks.

## 2 PROBLEM FORMULATION

**Online Decision-Making Problem.** As seen in Figure 1, each online decision-making phase begins with new articles (actions) $\mathcal{A}^{\text{new}}$ being released. Each article $a \in \mathcal{A}^{\text{new}}$ is associated with news article text (and possibly images) $Z^{(a)}$. These differ from "context" variables, which are common in bandit problems; here, $Z^{(a)}$ is associated with the action.[1] The system interacts sequentially with distinct

---

[1]Throughout we focus on the setting without context; we extend to settings with context in Appendix B.

users $t \in \{1, 2, \ldots, T\}$ and can adapt future recommendations based on previous user feedback. To the $t^{\text{th}}$ user, it recommends $A_t \in \mathcal{A}^{\text{new}}$, and associates with this a reward $R_t \in [0, 1]$. In particular, each action $a$ has $T$ potential rewards $R_{1:T}^{(a)} = (R_1^{(a)}, ..., R_T^{(a)})$. The observed reward is $R_t \leftarrow R_t^{(A_t)}$ if article $A_t$ is recommended to the $t^{\text{th}}$ user. Our goal is to develop a decision-making algorithm $\pi$ that maximizes the total expected reward, or equivalently minimize regret (formalized in Section 4).

**Historical Data.** We assume we have access to historical data $\mathcal{D}^{\text{hist}} = \{Z^{(a)}, R_{1:T}^{(a)}\}_{a \in \mathcal{A}^{\text{hist}}}$ on previously released news articles $\mathcal{A}^{\text{hist}}$. We use $\mathcal{D}^{\text{hist}}$ to pre-train our sequence models. We assume that the articles $\mathcal{A}^{\text{hist}}$ and $\mathcal{A}^{\text{new}}$ come from the same data generating process (formalized below). In real data settings, we may only have access to short sequences $R_{1:n}^{(a)}$ for some $n \leq T$; we discuss training techniques to get around this in practice in Section 6.

**Data Generating Process.** We model articles as independent draws from an unknown distribution $p^*$: $\{R_{1:T}^{(a)}, Z^{(a)}\} \sim p^*$ independently across $a \in \mathcal{A}^{\text{hist}} \cup \mathcal{A}^{\text{new}}$. We require $p^*$ to be *exchangeable*:

**Assumption 1** (Exchangeable). *$p^*$ is an exchangeable sequence model, i.e., for any $z$, for $R_{1:T}^{(a)} \sim p^*(\cdot \mid Z^{(a)} = z)$ the following are equal in distribution for any permutation $\sigma$ over $T$ elements:*

$$\left(R_1^{(a)}, \ldots, R_T^{(a)}\right) \mid \left(Z^{(a)} = z\right) \quad \overset{D}{=} \quad \left(R_{\sigma(1)}^{(a)}, \ldots, R_{\sigma(T)}^{(a)}\right) \mid \left(Z^{(a)} = z\right).$$

Assumption 1 ensures that the rewards $R_{1:T}^{(a)}$ are permutation invariant. A key running example of exchangeable sequence models $p^*$ are those associated with a conditionally i.i.d. data generation process ($p^*$ can be computed via Bayes rule). One simple example of this is the case of a Bayesian mixture model with a prior on some latent variable $U^{(a)}$, where

$$U^{(a)} \sim P(U^{(a)} \in \cdot \mid Z^{(a)}) \quad \text{then,} \quad R_1^{(a)}, \ldots, R_T^{(a)} \mid U^{(a)} \overset{i.i.d.}{\sim} P(R_t^{(a)} \in \cdot \mid U^{(a)}, Z^{(a)}). \quad (1)$$

The $p^*$ sequence model associated with the above data generating process satisfies Assumption 1.

## 3 POSTERIOR SAMPLING VIA AUTOREGRESSIVE GENERATION

Classical Thompson Sampling (TS) samples from a posterior distribution over mean rewards for each action and selects the action with the highest sample (Russo et al., 2020). TS requires the algorithm designer to (i) specify a meaningful belief (prior) over latent model parameters (e.g., unknown mean rewards for each action) and (ii) update that belief as the algorithm collects data. Both (i) and (ii) can be significant challenges when using neural networks (Riquelme et al., 2018).

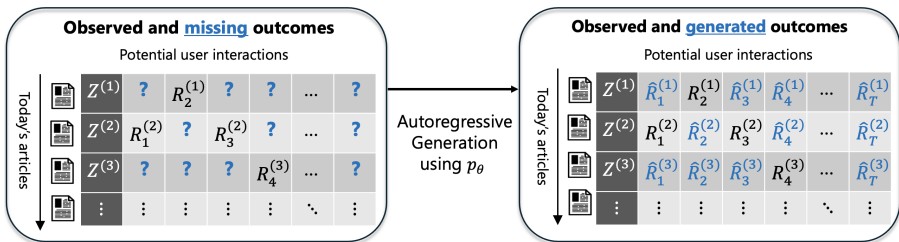

Figure 2: **Missing data viewpoint.** We view uncertainty about unobserved outcomes as the source of uncertainty, avoiding explicit reference to latent parameters or variables. Calibrated generation (imputation) of missing outcomes enables uncertainty quantification and decision-making.

In this work we implement Thompson sampling (TS) without explicitly modeling latent parameters. The impetus of our approach is that *unobserved rewards* are the source of the decision-maker's uncertainty (Figure 2): feedback on an article has only been gathered from a subset of users, and there is residual uncertainty in how future users would respond. Inspired by this viewpoint, our method proceeds in two steps. First, we pretrain an autoregressive sequence model to predict successive rewards using historical data $\mathcal{D}^{\text{hist}}$. Then, at decision time the algorithm uses imputed values of the missing rewards ($\hat{R}$'s) generated autoregressively from the pretrained sequence model to make decisions. We show this procedure is an principled implementation of TS. Our main insight applies to general autoregressive sequence models (e.g., transformers), but works well empirically even with simpler sequence model architectures (Section 6).

### 3.1 KEY INSIGHTS: CONNECTING SEQUENCE MODELLING AND BAYESIAN INFERENCE

What formal connection is there between sequence modeling and uncertainty quantification? It turns out there are two ideas in Bayesian inference that closely connect these two concepts.

**(1) Standard Bayesian models specify (exchangeable) autoregressive sequence models.** Bayesian models akin to those from equation 1 (i.e., with a prior on an unobserved latent variable and rewards that are i.i.d. given the latent) each uniquely define an exchangeable, autoregressive sequence model. Specifically, this sequence model is called the *posterior predictive* distribution (Gelman et al., 2013) and can be computed by integrating over the latent variable. This model specifies the distribution of $R_t^{(a)}$ given $Z^{(a)}, R_{1:t-1}^{(a)}$, which for the true data generating process we denote using $p^*(R_t^{(a)} \in \cdot \mid Z^{(a)}, R_{1:t-1}^{(a)})$ (recall we introduced this notation in Section 2).

**(2) One can sample from the posterior distribution of the mean reward by autoregressively generating from the posterior predictive.** In the Bayesian mixture model setting from equation 1, if we only had access to the posterior predictive distribution $p^*$, how can we obtain samples of the posterior for the expected reward $\mu^{(a)} := \int_r r \cdot P(R_t^{(a)} = r \mid U^{(a)}, Z^{(a)}) dr$ for each action $a$, which are needed for Thompson Sampling? It turns out approximate posterior samples can be obtained by autoregressively generating from the posterior predictive distribution $p^*$.

We form an approximate sample of $\mu^{(a)}$ from its posterior distribution given $R_{1:t-1}^{(a)}$. Sample $\hat{R}_t^{(a)} \sim p^*(R_t^{(a)} \in \cdot \mid Z^{(a)}, R_{1:t-1}^{(a)})$, $\hat{R}_{t+1}^{(a)} \sim p^*(R_{t+1}^{(a)} \in \cdot \mid Z^{(a)}, R_{1:t-1}^{(a)}, \hat{R}_t^{(a)})$, and so on until $\hat{R}_T \sim p^*(R_T^{(a)} \in \cdot \mid Z^{(a)}, R_{1:t-1}^{(a)}, \hat{R}_{t+1:T-1}^{(a)})$. Then, form the approximate posterior sample $\hat{\mu}_T^{(a)} := \frac{1}{T} \left\{ \sum_{k=1}^{t-1} R_k^{(a)} + \sum_{k=t}^{T} \hat{R}_k^{(a)} \right\}$. $\hat{\mu}_T^{(a)}$ is an exact posterior draw for the mean of the underlying $T$ reward potential outcomes $\mu_T^{(a)} := \frac{1}{T} \sum_{t=1}^{T} R_t^{(a)}$. Specifically, the following are equal in distribution for any $z$ and $r_{1:t}$ (since $\hat{R}_{t:T}^{(a)}$ and $R_{t:T}^{(a)}$ are drawn from the same distribution $p^*$):

$$\hat{\mu}_T^{(a)} \mid (Z^{(a)} = z, R_{1:t-1}^{(a)} = r_{1:t-1}) \quad \overset{D}{=} \quad \mu_T^{(a)} \mid (Z^{(a)} = z, R_{1:t-1}^{(a)} = r_{1:t-1}).$$

For large $T$, $\mu_T^{(a)}$ is close to the latent mean $\mu^{(a)} = \int r P(R_t^{(a)} = r \mid U^{(a)}, Z^{(a)}) dr$ (Appendix D.1).

The connection between autoregressive sampling and Bayesian inference rests on a link between exchangeable sequence modeling and Bayesian inference that has been known since de Finetti's seminal work (Finetti, 1933), and has appeared in several different literatures (Berti et al., 1998; Fortini et al., 2000; Fortini and Petrone, 2014; Hahn et al., 2018; Berti et al., 2021; 2022; Fong et al., 2023; Lee et al., 2023a). This is also related to Bayesian methods that impute missing outcomes by sampling from a posterior predictive distribution (Rubin, 1987; Gelman et al., 2013).

### 3.2 OUR ALGORITHM: POSTERIOR SAMPLING VIA AUTOREGRESSIVE GENERATION

The connection between Bayesian inference and sequence modeling suggests that one could implement Thompson sampling in complex settings without closed form posteriors, if one had access to the true posterior predictive distribution. This leads to several natural questions:

1. **How can we specify/learn good posterior predictive models when using neural networks?** We learn approximate posterior predictive distributions using autoregressive sequence models by training neural networks models to minimize a sequence prediction loss via gradient descent.

2. **What if the learned posterior predictive distribution is imperfect? What if it is not exactly exchangeable? How does that impact decision-making performance?** In practice, it is difficult to ensure that neural network-based sequence models are exactly exchangeable. We prove that the regret of an autoregressive generation version of Thompson sampling is controlled by how well the pretrained sequence model minimizes (the expected analogue of) the sequence loss it is trained on from equation 2, (regardless of whether the model is exactly exchangeable).

**Phase 1: Pretraining an Autoregressive Model.** We train an autoregressive sequence model $p_\theta$, parameterized by $\theta \in \Theta$, that can predict missing rewards, conditioned on article text, and limited previously observed rewards. This will enable us to generate hypothetical completions of the potential outcome table in Figure 2. Formally, this model specifies a probability $p_\theta(R_t^{(a)} \mid Z^{(a)}, R_{1:t-1}^{(a)})$

Figure 3: **Posterior Sampling via Autoregressive Generation (PS-AR).** PS-AR uses autoregressive generation (imputation) of unobserved potential outcomes to (implicitly) reason about uncertainty and drive exploration of arms that could plausibly be optimal. After imputing all missing rewards, the algorithm fits an optimal policy and executes the best action according to that policy.

of observing outcome $R_t^{(a)}$ from the next interaction conditioned on article attributes $Z^{(a)}$ and previous outcomes $R_{1:t-1}^{(a)}$. These one-step conditional probabilities generate a probability distribution over sequences as $p_\theta(R_{t:T}^{(a)} \mid Z^{(a)}, R_{1:t-1}^{(a)}) = \prod_{k=0}^{T} p_\theta(R_{t+k}^{(a)} \mid Z^{(a)}, R_{1:t+k-1}^{(a)})$. We use historical data $\mathcal{D}^{\text{hist}}$ to minimize the following loss function:

$$\ell(p_\theta; \mathcal{D}^{\text{hist}}) := - \sum_{a \in \mathcal{A}^{\text{hist}}} \sum_{t=1}^{T} \log p_\theta\big(R_t^{(a)} \mid Z^{(a)}, R_{1:t-1}^{(a)}\big). \tag{2}$$

In our experiments, we use bootstrap resampling to help ensure the sequence model is approximately exchangeable, reflecting Assumption 1. Our approach to pretraining approximately exchangeable sequence models closely mirrors recent work on neural processes (Garnelo et al., 2018b; Jha et al., 2022; Nguyen and Grover, 2022; Lee et al., 2023a) and prior-data fitted networks (Müller et al., 2022x). Our main contribution is linking this pretrained sequence model to online decision-making.

---

**Algorithm 1** Posterior Sampling via Autoregressive Generation (PS-AR)

---

**Require:** Autoregressive generative model $p_\theta$, actions $\mathcal{A}^{\text{new}}$ with $\{Z^{(a)}\}_{a \in \mathcal{A}^{\text{new}}}$
 1: Initialize list of missing entries $M^{(a)} \leftarrow [1, \dots, T]$ for each $a \in \mathcal{A}^{\text{new}}$
 2: **for** $t = 1, \dots, T$ **do**
 3:     **for** $a \in \mathcal{A}^{\text{new}}$ **do**
 4:         **for** $\tau \in M^{(a)}$ **do**
 5:             Sample missing reward: $\hat{R}_\tau^{(a)} \sim p_\theta\big(\cdot \mid Z^{(a)}, \big(R_i^{(a)}\big)_{i \notin M^{(a)}}, \big(\hat{R}_i^{(a)}\big)_{i \in M^{(a)}}\big)$
 6:         **end for**
 7:         Form imputed average reward: $\hat{\mu}_t^{(a)} \leftarrow \frac{1}{T}\big\{\sum_{\tau \notin M^{(a)}} R_\tau^{(a)} + \sum_{\tau \in M^{(a)}} \hat{R}_\tau^{(a)}\big\}$
 8:     **end for**
 9:     Select action $A_t \leftarrow \arg\max_{a \in \mathcal{A}^{\text{new}}} \big\{\hat{\mu}_t^{(a)}\big\}$ (break ties deterministically)
10:     Remove $t$ from the list of missing entries $M^{(A_t)}$
11:     Observe reward $R_t \leftarrow R_t^{(A_t)}$ from action $A_t$.
12: **end for**

---

**Phase 2: Online Decision-Making via Autoregressive Generation.** After a sequence model $p_\theta$ is trained on historical data, it is deployed and used for decision-making. No additional training of $p_\theta$ is needed. At each decision time, our algorithm uses $p_\theta$ to autoregressively generate (impute) missing rewards for each candidate action $a \in \mathcal{A}^{\text{new}}$ (Algorithm 1). Our algorithm then uses both the observed and generated rewards to fit an optimal policy and selects the best action according to that policy. In this simple setting without context, an optimal policy consists of taking a mean of the rewards for each arm to form $\hat{\mu}_t^{(a)}$ and selecting $A_t \leftarrow \arg\max_{a \in \mathcal{A}^{\text{new}}} \big\{\hat{\mu}_t^{(a)}\big\}$. Through this process, actions that are optimal under some likely generation of the missing rewards according to $p_\theta$ have a chance of being selected. Once no plausible sample of missing rewards could result in an action being optimal, it is essentially written off. Good performance of the algorithm relies on the model $p_\theta$ matching the data generating process closely (Section 4). We formally prove that our algorithm is an implementation of Thompson sampling in Appendix D.3.

**Disadvantages of Alternative Approaches to Using Sequence Models to Approximate Posterior Draws.** In line 7 of Algorithm 1, we average the imputed and observed rewards to form an approximate posterior draw. Alternative approaches of sampling from sequence models to approximate Thompson Sampling easily result in poor decision-making by over- or under-exploring (Figure 4).

Several works (Nguyen and Grover, 2022; Müller et al., 2022x; Garnelo et al., 2018a) propose choosing actions using the single-step *predictive uncertainty* in the next outcome (no averaging across users); while this works well when rewards are non-random, this reduces to random selection when rewards are random, as is the case in most real-world problems.

On the other hand, averaging across many independent (non-autoregressive) draws of the next outcome reduces to the mean of the predictive distribution and results in playing the action currently believed to be best, without purposeful exploration. Similar limitations apply if one uses the most likely sequence of outcomes, instead of sampling them randomly.

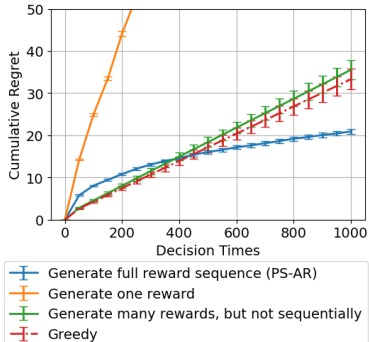

Figure 4: **Comparing to Alternative Sampling Approaches.** All methods use the same $p_\theta$ model, but PS-AR has the lowest regret at longer horizons: Appendix E.2.

**Advantages of the Autoregressive Approach to Thompson Sampling (TS).** In Table 1 we compare traditional TS with our generative version of TS (PS-AR). TS traditionally requires specifying a Bayesian model for latent parameters and performing explicit Bayesian inference over them; with neural networks this often involves making simplifying modeling assumptions, expensive Markov chain Monte Carlo, or heuristic posterior approximations. In contrast our autoregressive sequence modeling approach focuses on predicting missing rewards. This allows us to learn a Bayesian model from data using an easily measurable metric, i.e., the loss functions from equation 2.

Autoregressive sampling also aligns with emerging engineering practice. Pretraining using the loss from equation 2 requires learning a predictive model via loss minimization, as is standard practice. The PS-AR approach to uncertainty quantification can also take advantage of computational advances in autoregressive generation that are developed for other problem settings, e.g., LLMs.

|  | **Traditional Thompson Sampling** | **Generative Thompson Sampling** |
|---|---|---|
| Algorithmic Procedure | Sample latent model parameters, conditioned on observed rewards; optimize decisions with sampled latents | Probabilistically generate missing / future rewards; optimize decisions with generated rewards |
| Objects to Model (Bayesian) | Latent parameters and observed rewards (i.e., prior and likelihood) | Sequences of rewards (i.e., prior / posterior predictive distributions) |
| Learning the Bayesian Model (Pretraining) | Fit hyperparameters of prior and likelihood via empirical Bayes | Train an autoregressive sequence model to predict missing rewards |
| Online Decision-Making (Updating Beliefs Online) | Posterior inference (e.g., MCMC) | Condition on observed rewards in the sequence model |

Table 1: Comparison of "Traditional Thompson Sampling" vs "Generative Thompson Sampling"

### 3.3 INTERPRETING OUR PRE-TRAINING PROCEDURE AS EMPIRICAL BAYES

Empirical Bayes (Type-II maximum likelihood) is a method to fit Bayesian models (especially the prior distribution) to observed data (Murphy, 2022; Casella, 1985; Normand, 1999). Typically, Empirical Bayes is used to fit simple, Bayesian models with conjugate priors. It turns out that our pretraining procedure optimizes the same criterion used in Empirical Bayes.

For simplicity, consider a conjugate Bayesian model (without $Z^{(a)}$) where $\mu^{(a)} \sim \text{Beta}(\alpha, \beta)$ and $R_1^{(a)}, \ldots R_T^{(a)} \mid \mu^{(a)} \overset{i.i.d.}{\sim} \text{Bernoulli}(\mu^{(a)})$. Empirical Bayes fits the hyperparameters of the prior distribution, $(\alpha, \beta)$, by maximizing the *marginal likelihood* of the data $\mathcal{D}^{\text{hist}} = \{R_{1:T}^{(a)} : a \in \mathcal{A}^{\text{hist}}\}$:

$$P\big(\mathcal{D}^{\text{hist}}; (\alpha, \beta)\big) = \sum_{a \in \mathcal{A}^{\text{hist}}} \int_\mu \prod_{t=1}^T P\big(R_t^{(a)} \mid \mu^{(a)} = \mu\big) P\big(\mu^{(a)} = \mu; (\alpha, \beta)\big) d\mu.$$

Note that the marginal likelihood can also be decomposed via an autoregressive sequence criterion:

$$P\big(\mathcal{D}^{\text{hist}}; (\alpha, \beta)\big) = \sum_{a \in \mathcal{A}^{\text{hist}}} \prod_{t=1}^T P\big(R_t^{(a)} \mid R_{1:t-1}^{(a)}; (\alpha, \beta)\big).$$

Note that maximizing the above criterion is eqiuvalent to minimizing our training loss from equation 2, when the sequence model $p_\theta$ equals $P(\cdot; (\alpha, \beta))$. We show in this Beta-Bernoulli setting that we are able to recover the true Bayesian prior by training on our sequence loss (Appendix E.3).

# 4 REGRET BOUND

In this section we show that the expected loss of the learned sequence model $p_\theta$ controls the decision-making performance of our algorithm, reducing a challenging sequential decision-making problem to a loss minimization problem. Concretely, we establish a strong regret bound for PS-AR that depends on the expected loss achieved by $p_\theta$. The expected analogue of the loss from equation 2 (i.e., averaged over the draw of news articles) is

$$\ell(p_\theta) := \mathbb{E}\left[ -\sum_{t=1}^{T} \log p_\theta\big(R_t^{(a)} \mid Z^{(a)}, R_{1:t-1}^{(a)}\big) \right]. \tag{3}$$

We bound the expected per-user regret:

$$\Delta(\pi; p^*) := \mathbb{E}_{p^*, \pi}\left[ \max_{a \in \mathcal{A}^{\text{new}}} \left\{ \frac{1}{T} \sum_{t=1}^{T} R_t^{(a)} \right\} - \frac{1}{T} \sum_{t=1}^{T} R_t^{(A_t)} \right]. \tag{4}$$

In equation 4 above, we calculate the gap in reward relative to a oracle that always recommends the action with best performance in the population.[2] The above is a Bayesian regret, the expectation is taken over draws of potential outcomes $\{Z^{(a)}, R_{1:T}^{(a)}\}_{a \in \mathcal{A}^{\text{new}}}$, in addition any randomness in the $\pi$ itself. Because of the recurring nature of our problem (depicted in Figure 1), the expectation has a physical rather than philosophical meaning: $\Delta(\pi)$ is the long-run average regret the system would incur if $\pi$ were deployed across many days (and hence across many instances of the bandit task).

**Proposition 1.** *Under Assumption 1, for PS-AR (Algorithm 1) applied with $p_\theta$ (denoted $\pi_{\text{PS-AR}}(p_\theta)$),*

$$\Delta\big(\pi_{\text{PS-AR}}(p_\theta);\, p^*\big) \leq \underbrace{\sqrt{\frac{|\mathcal{A}^{\text{new}}| \log(|\mathcal{A}^{\text{new}}|)}{2T}}}_{\text{Regret bound for Thompson sampling}} + \underbrace{\sqrt{\frac{|\mathcal{A}^{\text{new}}|}{2}\big\{\ell(p_\theta) - \ell(p^*)\big\}}}_{\text{Penalty for sub-optimal prediction}}.$$

Wen et al. (2021) prove a regret bound that looks similar to ours, however their result requires a specific conditional KL divergence to be small, which does not appear to follow from training a model with low validation loss. Moreover we use a very different proof technique. Our Proposition 1 relies on Theorem 1, which is a result that may be of independent interest.

Theorem 1 uses an information-theoretic approach to show that when the distributions $p_\theta$ and $p^*$ are nearly indistinguishable in a Neyman-Pearson sense (i.e., the expected log likelihood ratio $\ell_n(p_\theta) - \ell_n(p^*)$ is small), *any* function of the potential outcomes generated under $p_\theta$ vs. $p^*$ must also be nearly indistinguishable. Below we use $\mathbb{E}_{p_\theta}$ to denote expectations under the distribution where the potential outcomes $R_{1:T}^{(a)}$ are generated autoregressively from $p_\theta$, i.e., $R_{1:T}^{(a)} \mid Z^{(a)} \sim p_\theta(\cdot \mid Z^{(a)})$.

**Theorem 1.** *Let $O^{\text{new}} := \big\{Z^{(a)}, R_{1:T}^{(a)}\big\}_{a \in \mathcal{A}^{\text{new}}}$ denote the potential outcomes table. Independent of $O^{\text{new}}$, let $\xi \sim \text{Uniform}[0, 1]$. Under Assumption 1, for real-valued functions $f$ of $O^{\text{new}}$ and $\xi$,*

$$\sup_{f:\|f\|_\infty \leq 1} \Big| \underbrace{\mathbb{E}_{p^*}\big[f\left(O^{\text{new}}, \xi\right)\big]}_{\text{Real Distribution}} - \underbrace{\mathbb{E}_{p_\theta}\big[f\left(O^{\text{new}}, \xi\right)\big]}_{\text{Simulated Distribution}} \Big| \leq \underbrace{\sqrt{(|\mathcal{A}^{\text{new}}|/2)\left\{\ell(p_\theta) - \ell(p^*)\right\}}}_{\text{Penalty for sub-optimal simulator}}.$$

We can apply Theorem 1 to show equation 5 because the per-user regret of an algorithm $\pi$ is simply a bounded function of all possible potential outcomes $O^{\text{new}}$ and exogenous noise $\xi$ if $\pi$ is a randomized algorithm. We formalize and prove this statement in Appendix D.5.

**Bounding the Regret of *any* Bandit Algorithm via a Simulator.** A consequence of Theorem 1 is that we can use it to bound the deployment regret of *any* policy $\pi$, i.e., $\Delta\big(\pi; p^*\big)$, in terms of the

---

[2]The difference between this oracle and one that selects the best expected reward is small (Appendix C).

regret under the simulator, $\Delta(\pi; p_\theta)$, and the gap in prediction loss $\ell(p_\theta) - \ell(p^*)$:

$$\underbrace{\Delta(\pi; p^*)}_{\text{Deployment regret}} \leq \underbrace{\Delta(\pi; p_\theta)}_{\text{Regret under simulator}} + \underbrace{\sqrt{(|\mathcal{A}^{\text{new}}|/2)\{\ell(p_\theta) - \ell(p^*)\}}}_{\text{Penalty for sub-optimal simulator}}. \tag{5}$$

Equation 5 says that the regret achieved by $\pi$ under the reward simulator $p_\theta$ is close to the regret of $\pi$ when deployed in the true environment $p^*$, so long as the prediction loss under $p_\theta$ and $p^*$ is close.

**Regret Bounds for Thompson Sampling (TS) with Misspecified Priors.** The secondary consequence of Theorem 1 is that it characterizes the regret of TS algorithms with misspecified priors. To see this, pick $\pi$ to be TS with a misspecified prior and pick $p_\theta$ to be the data generating distribution under the misspecified prior; then $\Delta(\pi; p_\theta)$ will have the typical regret bound for TS and the second term on the RHS of equation 5 characterizes the penalty for having a misspecified prior. Our result builds on previous literature which prove lower bounds for TS under misspecified priors (Liu and Li, 2016) and upper bound the regret of a "k-shot" version of TS (Simchowitz et al., 2021).

## 5 RELATED WORK

**Meta-Learning in Bandits.** There are a variety of meta-learning bandit algorithms (Wan et al., 2023; Cella et al., 2020; Kveton et al., 2021; Bastani et al., 2022); these methods primarily focus on simpler settings (e.g. Gaussian or linear reward models). There are also deep meta-learning methods developed for recommendation systems and the cold-start problem (Wang et al., 2022; Zhang et al., 2021; Zheng et al., 2021). These works primarily focus on more complex recommendation settings (e.g. tracking one user over time) and not on uncertainty. In contrast, our goal is to showcase our uncertainty quantification method for decision making with foundation models.

**Reinforcement Learning (RL) with Pre-Trained Autoregressive Models.** Many recent works in RL leverage sequence models that are pretrained on a large volume of data collected by an expert policy. Some use goal-conditioned sampling of actions to improve over average expert behavior (Janner et al., 2021; Chen et al., 2021; Ding et al., 2019); this works well in some settings but is provably sub-optimal others (Brandfonbrener et al., 2022; Malenica and Murphy, 2023). In contrast to these works, we do not require data collected by expert policies for training.

Decision Pretrained Transformers (DPT) (Lee et al., 2023b) relate sampling from a sequence model that predicts the next expert action to Thompson sampling. Similar to PS-AR, DPT focuses on a meta-learning setting and pretrains their sequence models on historical data. However, while PS-AR sequence models are trained to predict future rewards, DPT sequence models are trained to predict the next expert action (if an expert action is not available, it is trained to mimic an approximate optimal policy fit from data). We have preliminary experiments comparing to DPT (Appendix E.6) and find that it performs similarly to PS-AR; further investigation is needed to understand the benefits of predicting future rewards versus expert actions.

Autoregressive predictive models are also used in Liu et al. (2023), however they focus on non-stationary environments and settings with closed-form posterior distributions (like AR processes); they do not discuss incorporating deep learning models or characterizing model performance when the sequence model is misspecified.

**Thompson Sampling with Neural Networks.** Several classes of approaches that have emerged to scale Thompson sampling to modern large scale decision-making problems with neural networks. The first class places a Bayesian prior on the weights of the neural network itself. These methods include those that form a Bayesian linear regression model from the last layer of a trained neural network (Riquelme et al., 2018; Snoek et al., 2015), as well as Bayesian neural networks (Zhang et al., 2020). A second class of approaches involves forming using an ensemble of neural networks to simulate samples from a posterior distribution (Osband et al., 2018; Lu and Van Roy, 2017; Qin et al., 2022a). This class also includes algorithms that build on Epinets (Osband et al., 2024; Zhu and Van Roy, 2023; Osband et al., 2023) and HyperModels (Dwaracherla et al., 2020; Li et al.), which attempt to retain the performance of the ensembling with lower computational cost. Notably, Osband et al. (2024) uses sequence prediction loss to *evaluate* the quality of ("epistemic") uncertainty quantification, inspiring our efforts to construct bandit algorithms using sequence models.

# 6 EXPERIMENTS

We evaluate our approach in a synthetic setting and in a semi-realistic news recommendation setting. While our method applies more broadly, we focus on binary rewards ($R_t \in \{0, 1\}$) as the news dataset we build on has binary click/no click outcomes. We first discuss implementation techniques:

**(1) Bootstrapping Training Data.** In practice, the historical $\mathcal{D}^{\text{hist}}$ may have sequences $R_{1:n}^{(a)}$ for some $n$ less than the horizon $T$. To ensure the learned sequence model $p_\theta$ has low prediction loss, $\ell(p_\theta)$, for longer sequences, we bootstrap the data in training by computing the loss on bootstrapped sequences of rewards sampled with replacement (see Appendix A for details). This procedure also helps ensure the sequence model is approximately exchangeable, reflecting Assumption 1.

**(2) Truncating Generation Lengths.** When the population size $T$ is large, generating missing outcomes for the entire population can be costly. To save computation, we implement a slightly modified version of PS-AR that instead generates only $m$ missing outcomes per action and averages those $m$ outcomes to form $\hat{\mu}_t^{(a)}$. We find as long as $m$ is sufficiently large, truncation makes little difference in practice (see Figure 8).

## 6.1 SYNTHETIC SETTING: MIXTURE BETA-BERNOULLI

Our synthetic experiments use a mixture model where $Z^{(a)} \in \mathbb{R}^2$ and the prior is a mixture of two Betas and the likelihood is Bernoulli. See Appendix E.1 for more details.

**Models.** We consider two sequence model $p_\theta$ variants. (i) FLEXIBLE NN is a neural network that takes $Z^{(a)}$ and a summary of the past outcomes for action $a$ as input. (ii) BETA-BERNOULLI NN, is the closed-form posterior predictive for the Beta-Bernoulli model; its hyperparameters $\alpha_\theta(Z^{(a)})$ and $\beta_\theta(Z^{(a)})$ are parameterized by neural networks that take $Z^{(a)}$ as input.

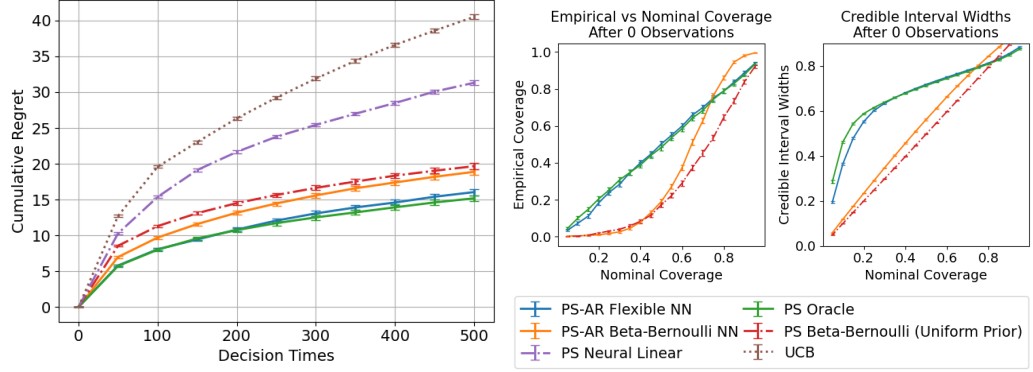

Figure 5: **Evaluation in mixture Beta-Bernoulli Setting.** Left: cumulative regret with $|\mathcal{A}^{\text{new}}| = 10$, averaged over 500 repetitions. Right: evaluating uncertainty quantification (coverage and interval width) averaged over 1000 actions not seen in training. Error bars are $\pm 1$ s.e.

**Regret: Figure 5 (Left).** PS ORACLE, which implements Thompson (posterior) sampling with a prior that matches the data generating process, has the lowest regret. PS-AR FLEXIBLE NN closely matches the performance of PS ORACLE. PS-AR BETA-BERNOULLI NN which uses a sequence model with a misspecified, unimodal Beta prior performs similarly to PS BETA-BERNOULLI (UNIFORM PRIOR) which performs exact Thompson sampling with a uniform prior. These Thompson sampling-based algorithms outperform the UCB algorithm (Abbasi-Yadkori et al., 2011) PS NEURAL LINEAR, Thompson sampling with a linear Gaussian bayesian model with an uninformative prior on top of learned text embeddings, and DPT, a sequence model trained to predict best action from a set of histories (Lee et al., 2023b). See more on baseline algorithms in Appendix E.5.

**Uncertainty Quantification: Figure 5 (Right).** For 1000 articles not seen in training, we form 250 posterior samples $\hat{\mu}_1^{(a)}$ by autoregressively generating outcomes conditional on $Z^{(a)}$ using $p_\theta$. We use the percentiles of the sampled $\hat{\mu}_1^{(a)}$'s to form credible intervals and evaluate how often the true $\mu_1^{(a)}$ is within these intervals. The intervals generated by the FLEXIBLE NN sequence model

have excellent coverage; moreover, the width of the intervals are the narrowest that have correct coverage (matching PS ORACLE). In contrast, the BETA-BERNOULLI NN sequence model which has a unimodal (misspecified) Beta prior has worse coverage.

## 6.2 NEWS RECOMMENDATION SETTING

We build a semi-realistic news recommendation task using the MIcrosoft News Dataset (MIND) (Wu et al., 2020). This setting demonstrates how PS-AR easily integrates with pretrained language models. Here $Z^{(a)}$ is article headline text or news category information (e.g. "politics" or "sports"). Rewards are binary click/no-click outcomes. After pre-processing, the dataset has $\approx$ 11k articles.

**Models** We use three $p_\theta$ model variants: (i) FLEXIBLE NN (TEXT) and (ii) BETA-BERNOULLI NN (TEXT), are analogous to those from Section 6.1, but we modify them to use article text $Z^{(a)}$ embedded using DistilBERT (Sanh et al., 2019), which is fine-tuned end-to-end during pretraining. (iii) FLEXIBLE NN (CATEGORY) the final model uses category information instead of headline text.

**Regret and Uncertainty Quantification: Figure 6** In terms of regret, the PS-AR models that use sequence models $p_\theta$ that incorporate text features outperform all other algorithms (baselines described in Section 6.1). We use an analogous procedure as used in Section 6.1 to form uncertainty intervals for $\mu_1^{(a)}$ for the 2280 actions not seen in training. All PS-AR models have intervals with correct coverage, but the text-based models have slightly narrower intervals. We also compare to an ensemble of 50 models, which we found has poor coverage. See Appendix E.4 for more details.

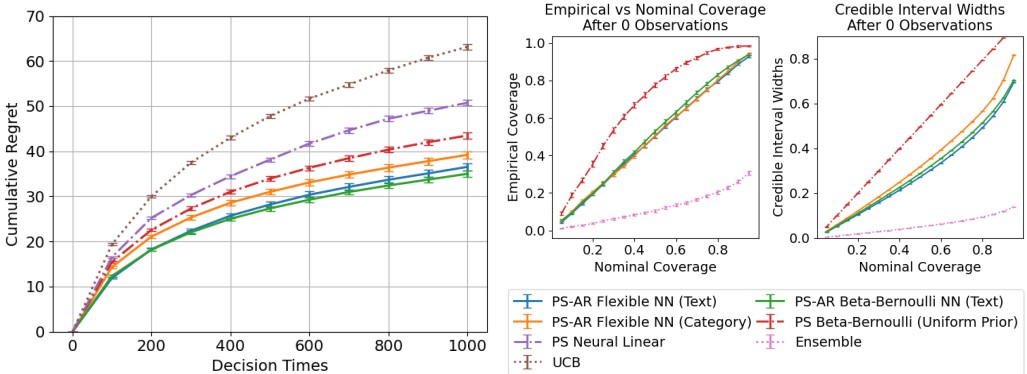

Figure 6: **Evaluation on news data.** Left: cumulative regret with $|\mathcal{A}^{\text{new}}| = 10$, averaged over 500 repetitions. Right: evaluating uncertainty quantification (coverage and interval width), averaged over 2280 actions not seen in training. Error bars are $\pm 1$ s.e.

## 7 DISCUSSION

We formulate a loss minimization problem that implicitly learns an informed prior using historical data, in order to model the posterior distribution of rewards for decision-making. This connection enables using modern ML tools to learn rich representations to comprehend uncertainty, in an actionable way. Our formulation introduces a fresh approach to the longstanding challenge of scaling Thompson sampling to incorporate neural networks that incorporate unstructured inputs such as images and text (Riquelme et al., 2018). The main ideas behind our algorithm generalize to *contextual* settings where user-specific contexts $X_t$ can be used to tailor recommendation decisions. We describe generalizing PS-AR to this setting in Appendix B and leave a deeper dive to future work.

**Limitations.** We assume articles are i.i.d. between pretraining and online evaluation, and user outcomes for each action are exchangeable. Such assumptions may not be appropriate in practice, e.g., if user preferences are nonstationary. In conducting this work, we struggled to find publicly available datasets on which to evaluate our method, which led us to build our news recommendation setting. Building public benchmarks for bandit problems that require using complex inputs (e.g. text and/or images) for best performance is an important open direction. A limitation of this work is we do not provide a thorough answer as to the quality of the historical data (e.g., amount of data and/or how data was collected) necessary to ensure learning good sequence models.

**Reproducibility Statement.** Full details for reproducing the experiments, including data processing, are in Appendix E, with code in the supplemental materials. For theoretical results in Section 4, the proofs are in Appendix D.

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
