# OpenReview forum: "Informed Exploration via Generative Modeling"
_ICLR.cc/2025/Conference — ICLR 2025 Conference Withdrawn Submission_

### Official Review · Reviewer_vLMe · 2024-11-02

**Soundness:** 3
**Presentation:** 2
**Contribution:** 3
**Rating:** 5
**Confidence:** 3

**Summary:**

This paper studies exploration under general neural networks. They propose the method kind of like "generative Thompson Sampling", which does not explicitly model the unknown latent variables, but aims at modeling the missing rewards. The procedure of the algorithm starts with implicitly learns the Bayesian model via pre-training by minimizing a sequence prediction loss on successive reward using historical data, then at inference time, it uses the both the observed and imputed reward to make decisions. This basically reduces
 the sequential decision-making problem to a sequence prediction training with low loss. Experiments are done on synthetic and news recommendation dataset.

**Strengths:**

- This paper tackles the important issue of exploration in general neural networks, a challenge with broad implications for real-world applications like recommendation systems, education, and even LLM generation. Informed exploration are critical for these complex decision-making systems.
- The authors effectively bridge the gap between sequence modeling and Bayesian inference, introducing the concept of "generative Thompson sampling". This provides a strong theoretical justification for their approach and nicely clarifies the role of pre-training as a form of empirical Bayes.
- The proposed method is easy to implement, and the experimental results on recommendation tasks demonstrate the effectiveness of the proposed approach, in terms of both reducing regret and providing valid confidence intervals.

**Weaknesses:**

- Lack of Clarity and Potential Inconsistencies: The paper would benefit from increased clarity in its methodological details and problem formulation. Specifically:

1.  While the motivating example of news recommendation emphasizes no overlap in the action set across time steps, this constraint seems not to be the case  in the formal problem definition and Algorithm 1.
2. The role of different users in Equation 2 and Algorithm 1 remains unclear. It appears to estimate a population-level mean reward for each action, over all users. This then seems a multi-arm bandit problem than the contextual bandit one? The authors should explicitly state whether they are addressing a multi-arm bandit or a contextual bandit problem and revise the relevant sections accordingly. It is hard to estimate the context-dependent reward based on the current alg.

- Computational Cost: The proposed method's computational complexity scales linearly with both the action set size and the "overlap" between new and historical actions, and time-step T. This could become prohibitive in applications with large action spaces or significant small overlap, even disjoint support. The authors should discuss more on the scaling of the method.

- Limited Novelty: The proposed approach is very similar to reward models [1] employed in the offline bandit and RL literature. While subtle differences may exist, the authors need to provide a more thorough comparison with existing reward modeling techniques, explicitly highlighting their novel contributions here.

Ref: https://arxiv.org/pdf/1103.4601

**Questions:**

See weakness.

---

> ### Author Response · Authors · 2024-11-20
>
> We are glad that the reviewer finds our method intuitive and easy to implement, and that the reviewer appreciates that we tackle the important problem of informed exploration, which is crucial in many important settings. We are also glad that the reviewer appreciates our strong theoretical justification and connections to empirical Bayes, and that the reviewer recognizes that our experiments demonstrate effectiveness in terms of both regret and calibrated confidence intervals.
>
> ## 1. Overlap
> As discussed in line 114 in our problem setup, the articles in the historical dataset and the online decision-making phase are assumed to be drawn IID from the same distribution. Note that we use $\mathcal{A}^{\mathrm{hist}}$ throughout to denote the articles observed in the historical dataset, and use $\mathcal{A}^{\mathrm{eval}}$ to denote the articles observed in the online decision-making phase (and in Algorithm 1), which are new articles that are not in  $\mathcal{A}^{\mathrm{hist}}$.
>
> ## 2. Multi-arm vs contextual bandit setting
> For ease of exposition, the main text of our work focuses on the multi-arm bandit setting, without user context. There, we assume multiple users, but without user attributes. This is equivalent to viewing the problem as one of finding the best article, on average, for a cluster of users with similar interests or attributes.
>
> In Footnote 1 on page 2, we explain that we defer a discussion of the setting with user attributes ("context"), including algorithms, to Appendix B in the Supplementary Materials. We highlight this choice of presentation on line 108 and the footnote at the bottom of page 2 at the beginning of the problem formulation section.
>
> We will emphasize these points in the revision.
>
> ## 3. Computational cost
> * For the online decision-making phase, autoregressively generating from the sequence model can be computationally intense, and scales with the number of timesteps. We discuss truncating the generation length in our experiments (see line 443), which we found works well empirically. We find that if we completely remove autoregressive generation, the decision-making algorithm makes lower quality decisions (see Figure 4).
> * For both online and offline portions, computational cost for PSAR scales linearly with the number of online and offline actions considered, respectively. This is part is largely unavoidable. We note that the computational scaling of our approach (especially the offline pretraining) is similar to that of training and generating from LLMs. We expect that the cost of sequence modeling (both training and inference) will decrease, as is the current trend for LLMs. We highlight this on line 298, in the paragraph titled "Advantages of the Autoregressive Approach to Thompson
> Sampling (TS)".
> * We think there may be a misunderstanding regarding overlap; see the section of our response regarding overlap.
>
> ## 4. Related work
> The paper the reviewer suggested as a related work [1] is not relevant to our work. The suggested paper is on offline policy evaluation, while ours is on online decision-making. Because there is no online data gathering in [1], there is no need to balance exploration and exploitation, and thus the suggested paper has no relation to Bayesian uncertainty. While minor similarities may exist (e.g. modeling potential rewards under different actions, which is conceptually fundamental to all of bandit decision-making), the suggested work is not related.
>
> [1] Doubly Robust Policy Evaluation and Learning. Dudik, Langford, and Li. https://arxiv.org/pdf/1103.4601

---

### Official Review · Reviewer_UuKg · 2024-11-07

**Soundness:** 3
**Presentation:** 2
**Contribution:** 3
**Rating:** 5
**Confidence:** 3

**Summary:**

This paper aims to address the challenge of modeling uncertainty in decision-making in news recommendation systems. The authors propose a method that leverages autoregressive generative models to predict the missing or future rewards based on the historical, and then use the Thompson sampling for final decision-making with the previous and generated rewards. This paper proposes several theory analyses of the advantages of using sequential models (like transformers). The authors demonstrate their approach through both synthetic and semi-realistic news recommendation experiments, showing that their method can effectively incorporate pretrained language models and achieve strong performance in terms of regret minimization and uncertainty quantification.

**Strengths:**

1) The idea of incorporating sequential models into decision-making makes sense to me. Pre-trained on previous data, the autoregressive model has the ability to predict the missing reward values, which is benefit for the downstream algorithm.

2) The authors provide several theory analyses to support the efficiency of the proposed modules.

3) Experiments on both synthetic and real-world settings show the improvements of the proposed model.

**Weaknesses:**

1) One of the main concerns comes from the experiments. The authors only test their models with the base models (such as PS Neural Linear, PS Beta-Bernoulli), which may reduce the convincingness of the proposed method.

2) The authors claim that they use the language foundation model to improve the uncertainty quantification, which is not precise. Because they only pre-trained sequential models on historical data, the trained models are not actually foundation models.

3) The Thompson sampling at the inference stage relies heavily upon the generated missing reward values, which may limit its application when sequence models cannot precisely predict missing rewards.

**Questions:**

Please see the above Weaknesses section.

---

> ### Author Response · Authors · 2024-11-20
>
> We appreciate that the reviewer found our method intuitive, values our theoretical analysis, and values our experiments in both synthetic and real-world settings.
>
> ## 1. Choice of baselines
> Deep Bayesian Bandits Showdown [1] evaluates a variety of Bayesian decision-making algorithms when scaled with neural networks. In that work, NeuralLinear and NeuralGreedy were found to outperform a variety of other approaches (e.g. variational inference, Bayes by Backprop, Expectation propagation, dropout). Thus, in our work we chose to compare to neural linear and greedy (see Figure 4) in terms of regret, and also compare to ensembling in terms of uncertainty quantification. For these reasons, we believe the methods we compare to are strong baselines. We will emphasize these points in the revision.
>
> ## 2. Use of foundation models
> To clarify definitions, foundation models are models trained on large quantities of data that can be tailored (e.g. via fine-tuning) to more specific downstream tasks [2]. We use BERT, which a foundation model for text, and we fine-tune a sequence model that uses BERT in our setting. We feed text associated with a news article through BERT to get a text representation, which is fed into a sequence model used to predict future rewards; the entire model, including the BERT component, is fine-tuned end-to-end. We find that without (i) the LLM embedding, and (ii) without end-to-end fine tuning that the performance on the downstream decision-making problem goes down. We have described this procedure in the main text (see line 42), where we describe the architecture as "a neural network model with a large language model (LLM) component".
>
> ## 3. What if the sequence models cannot precisely predict outcomes?
> Our approach supposes that one is able to pretrain a sequence model to sample from a distribution of missing outcomes appropriately, and we discuss how standard sequence modeling approaches (training via standard loss minimization and gradient descent) can be used for this approach. In other words, techniques used to train sequence models to predict missing/masked out future words, can also be employed to predict missing/masked out future rewards.
>
> Furthermore, in our theory, we characterize a bound on the quality of the decision-making algorithm depending on the loss of the pretrained model; this provides a way to easily evaluate prior to deploying the algorithm, whether PSAR may be a suitable algorithm for the problem (which is a significant advantage that many other decision-making algorithms do not provide).
>
>
> [1] Riquelme, Carlos et al. “Deep Bayesian Bandits Showdown: An Empirical Comparison of Bayesian Deep Networks for Thompson Sampling.” ICLR 2018
>
> [2] Wikipedia page for Foundation models  https://en.wikipedia.org/wiki/Foundation_model

---

### Official Review · Reviewer_gvpA · 2024-11-09

**Soundness:** 2
**Presentation:** 4
**Contribution:** 2
**Rating:** 5
**Confidence:** 4

**Summary:**

The paper explores uncertainty in neural networks for decision-making, focusing on a meta-learning bandit setup applied to news recommendation. It introduces a novel approach using autoregressive generative modeling to quantify Bayesian uncertainty by predicting future rewards based on historical data. This method involves pre-training a generative model to forecast potential rewards, which helps balance exploration and exploitation without directly modeling latent variables. The approach is demonstrated through theoretical bounds and experiments, showing it enhances decision-making accuracy by effectively integrating neural networks.

**Strengths:**

* The paper presents a new method for Bayesian uncertainty quantification through autoregressive generative modeling, which provides a scalable and effective alternative to traditional methods that rely on complex posterior approximations.

* By applying this method to a news recommendation task, the paper demonstrates the approach's potential to improve real-world decision-making where uncertainty and user interactions play a crucial role.

* The paper provides a formal regret bound, showing that the approach’s decision-making performance directly relates to the sequence model’s training loss, solidifying the theoretical basis for the proposed method.

**Weaknesses:**

* A major concern is that this paper lacks a thorough discussion of related work, despite several highly relevant and overlapping studies. There is existing research that uses generative modeling for posterior approximations【1】and in recommendation tasks【2】. Additionally, multiple works have explored connections between autoregressive generative modeling and approximate Bayesian inference【3,4】. Furthermore, meta-learning with autoregressive generative networks has been studied in contexts such as classification【5】, Bayesian Optimization (BO)【6】, and decision-making in sequential prediction【7】.


1. Gal, Yarin, and Zoubin Ghahramani. "Dropout as a bayesian approximation: Representing model uncertainty in deep learning." international conference on machine learning. PMLR, 2016.
2. Da Tsai, Yun, and Shou De Lin. "Fast online inference for nonlinear contextual bandit based on generative adversarial network." arXiv preprint arXiv:2202.08867 (2022).
3. Hollmann, Noah, et al. "Tabpfn: A transformer that solves small tabular classification problems in a second." arXiv preprint arXiv:2207.01848 (2022).
4. Müller, Samuel, et al. "Transformers Can Do Bayesian Inference." International Conference on Learning Representations.
5. Bonet, David, et al. "HyperFast: Instant Classification for Tabular Data." Proceedings of the AAAI Conference on Artificial Intelligence. Vol. 38. No. 10. 2024.
6. Müller, Samuel, et al. "Pfns4bo: In-context learning for bayesian optimization." International Conference on Machine Learning. PMLR, 2023.
7. Lee, Jonathan, et al. "Supervised pretraining can learn in-context reinforcement learning." Advances in Neural Information Processing Systems 36 (2024).

**Questions:**

* Could the authors clarify the differences between their approach and existing works, highlighting the novelty and contribution of this work?
* Could the authors clarify if the proposed methods are used to address the delayed feedback issue (missing outcomes) rather than used for building a recommendation policy?

---

> ### Author Response · Authors · 2024-11-20
>
> We appreciate the positive comments about our novel approach to Bayesian uncertainty quantification that we use with decision-making algorithms, in a way that is scalable and effective, is demonstrated on real-world tasks, and is furthermore theoretically grounded.
>
> We thank the reviewer for their suggestions regarding related work on methods for uncertainty quantification. We emphasize that our main contribution is our **decision-making algorithm that quantifies uncertainty via sequence modeling, and which is a principled implementation of Thompson sampling**. Our contribution furthermore includes the empirical evaluation and theoretical analysis of our method. In the revision, we aim to emphasize the *decision-making* aspect of our contribution more clearly; see [8] for an overview of this challenge.
>
> We also want to further emphasize how our algorithm PSAR stands out:
> * PSAR can power efficient exploration in settings where there are rich unstructured attributes (e.g. text) associated with different actions. Other methods do not take advantage of action attributes.
> * Previous efforts to use sequence models to implement Thompson sampling, only sample the next reward, rather than a sequence of next rewards; these do not implement posterior sampling correctly when rewards are noisy and can have bad performance (line 270-275, Figure 4).
> * Our theory quantifies rigorously that effective next-token prediction is sufficient for interactive decision-making and exploration, a substantial result missing in all previous work.
>
> ### Comments and revisions based on suggested related work.
> Regarding the papers you listed, several of them are already mentioned in the original draft, while others we plan to add a discussion of:
> * Decision Pretrained Transformers [7]: We have already included a discussion of [7]; see line 410.
> * Prior Fitted Networks: We have a brief discussion of prior-fitted networks [4] and neural processes on lines 236-238. We plan to expand this discussion, and to include [3,6]. Specifically, we plan to say:
>     * _While prior fitted networks (PFNs) [3,4,6] develop generative models for Bayesian uncertainty quantification, except for [6], none of these works develop methods for on online decision-making. [6] focuses on a Bayesian optimization setting, while we consider a bandit decision-making setting where actions can have complex attributes (e.g. news article text). Furthermore, while we prove regret bound guarantees for our PSAR algorithm, [6] does not prove any performance guarantees for their approach. In general, we hypothesize that the work on PFNs in developing pretraining and architectural methods is very compatible with our PSAR decision-making algorithm, and believe using PSAR with PFNs is an interesting direction for future research._
> * In the revision we plan to discuss [2], which considers a contextual bandit algorithm that utilizes GANs. A primary disadvantage of their approach is that they require taking gradient steps in the generator and discriminator in the online decision-making phase, which is practically difficult in online decision-making problems. In contrast, our approach all the pretraining and gradient steps happen in the pretraining phase.
>
> We do not plan to discuss the following work, as the focus is too tangential to our contribution:
> * [1] discusses Bayesian uncertainty for neural networks via dropout, but not generative models for Bayesian uncertainty quantification. Furthermore, [10] which investigated a variety of Bayesian neural network methods with Thompson sampling found that dropout did not perform well. This motivated us to compare to Neural Linear in our simulations, which was among the best performing algorithms in [10].
> * [5] is a meta-learning method for fast classification of tabular data, similar to [3]. However, while [3] is (essentially) Bayesian, [5] is not. Thus, [5] is neither about generative models for Bayesian uncertainty quantification, nor online decision-making.
>
> We believe the current proposed revisions balance the need to acknowledge related contributions while preserving the clarity of our main arguments.
>
> ### Missing outcomes and decision-making
> We had some trouble understanding your question on missing outcomes. We do not consider a delayed reward setting, where outcomes are delayed by e.g. weeks or months, as in [9]. Rather, since we are in a bandit decision-making problem, the algorithm only observes rewards for the actions it selects, and the rewards associated with the unselected actions are always missing. The algorithm also generates rewards for future decision times, _within the same day_.
>
> As detailed in the paper and summarized in Algorithm 1, the imputed missing outcomes (during decision-making) are used to determine the action to take. In the simple multi-arm bandit setting, the arm with the highest (average) imputed missing outcomes is chosen. This is an implementation of Thompson sampling.

---

> > ### Author Response · Authors · 2024-11-20
> >
> > [1] Gal, Yarin, and Zoubin Ghahramani. "Dropout as a bayesian approximation: Representing model uncertainty in deep learning." ICML 2016
> >
> > [2] Da Tsai, Yun, and Shou De Lin. "Fast online inference for nonlinear contextual bandit based on generative adversarial network." arXiv preprint 2022
> >
> > [3] Hollmann, Noah, et al. "Tabpfn: A transformer that solves small tabular classification problems in a second." ICLR 2023
> >
> > [4] Müller, Samuel, et al. "Transformers Can Do Bayesian Inference." ICLR 2022
> >
> > [5] Bonet, David, et al. "HyperFast: Instant Classification for Tabular Data." AAAI 2024
> >
> > [6] Müller, Samuel, et al. "Pfns4bo: In-context learning for bayesian optimization." ICML 2023
> >
> > [7] Lee, Jonathan, et al. "Supervised pretraining can learn in-context reinforcement learning." NeurIPS 2024
> >
> > [8] Riquelme, Carlos et al. “Deep Bayesian Bandits Showdown: An Empirical Comparison of Bayesian Deep Networks for Thompson Sampling.” ICLR 2018
> >
> > [9] McDonald, Thomas et al. "Impatient Bandits: Optimizing Recommendations for the Long-Term Without Delay." KDD 2023

---

### Author Response · Authors · 2024-11-20

We thank the reviewers for their feedback. As the reviewers had different questions and concerns, we address those in each response separately. In the meantime, we emphasize our contribution and its significance:

**Our primary contribution is our scalable algorithm for decision-making under uncertainty.** The novelty and contribution in our work is in connecting sequential decision-making and generative sequence models for modeling unobserved rewards, in a theoretically grounded way. We have
1. Formulated and instantiated a meta-bandit problem with rich (unstructured) arm-features that requires interfacing with foundation models, that is common in industry (e.g. recommendation systems) but that does not yet exist in the literature,
2. Proposed a correct and novel implementation of Thompson sampling in this problem that uses auto-regressive generation of missing rewards to power exploration,
3. Demonstrated the effectiveness of this method on both synthetic and real-world data, and
4. Provided a novel theorem that controls the quality of interactive decision-making (i.e. regret) by the quality of offline prediction (near-optimality in terms of next-token prediction loss).

This combination of contributions is unique to our work.

---

### Note · Authors · 2024-12-04

**Comment:**

Due to misunderstandings of our work and the lack of feedback from reviewers, we have chosen to withdraw our paper from consideration.

**Withdrawal Confirmation:**

I have read and agree with the venue's withdrawal policy on behalf of myself and my co-authors.